# Knowledge Fusion of Large Language Models

**Fanqi Wan**[1]*, **Xinting Huang**[2]†, **Deng Cai**[2], **Xiaojun Quan**[1]†, **Wei Bi**[2], **Shuming Shi**[2]

[1]School of Computer Science and Engineering, Sun Yat-sen University, China
[2]Tencent AI Lab

wanfq@mail2.sysu.edu.cn, quanxj3@mail.sysu.edu.cn
{timxthuang,jcykcai,victoriabi,shumingshi}@tencent.com

## Abstract

While training large language models (LLMs) from scratch can generate models with distinct functionalities and strengths, it comes at significant costs and may result in redundant capabilities. Alternatively, a cost-effective and compelling approach is to merge existing pre-trained LLMs into a more potent model. However, due to the varying architectures of these LLMs, directly blending their weights is impractical. In this paper, we introduce the notion of knowledge fusion for LLMs, aimed at combining the capabilities of existing LLMs and transferring them into a single LLM. By leveraging the generative distributions of source LLMs, we externalize their collective knowledge and unique strengths, thereby potentially elevating the capabilities of the target model beyond those of any individual source LLM. We validate our approach using three popular LLMs with different architectures—Llama-2, MPT, and OpenLLaMA—across various benchmarks and tasks. Our findings confirm that the fusion of LLMs can improve the performance of the target model across a range of capabilities such as reasoning, commonsense, and code generation. Our code, model weights, and data are public at https://github.com/fanqiwan/FuseLLM.

## 1 Introduction

With the continuous success of large language models (LLMs) such as GPT (Brown et al., 2020) and LLaMA (Touvron et al., 2023) series across a wide range of natural language processing (NLP) tasks, it has become a strategic imperative for corporations to create their own LLMs. However, the costs associated with LLM development are astronomical. In addition to requiring vast amounts of training data, advanced techniques, substantial computational resources, and skilled labor, the development process also exerts significant pressure on energy consumption and the environment (Rillig et al., 2023). While these LLMs exhibit structural and functional differences, they share similar capabilities across a spectrum of NLP tasks. Consequently, beyond the traditional approach of training an LLM from scratch, an alternative option is to combine existing LLMs into a new, more powerful one, which is termed *knowledge fusion of LLMs* in this paper. If successful, this fusion not only cuts the cost of initial training but also allows the integrated model to benefit from the strengths of all the LLMs. This new model can also be fine-tuned and adapted for various downstream tasks. Moreover, the fusion can also happen among fine-tuned LLMs that specialize in a specific task.

The endeavor to integrate the capabilities of multiple models has been a long-standing pursuit. For example, ensemble methods (Littlestone & Warmuth, 1994; Jiang et al., 2023) directly aggregate the outputs of different models to enhance prediction performance and robustness. However, this approach requires maintaining multiple trained models and executing each during inference, which is impractical for LLMs due to their substantial memory and inference time requirements. Likewise, this approach doesn't facilitate fine-tuning, which is essential for many LLMs. Another approach is to directly merge several neural networks into a single network through parameter-wise arithmetic operations (Wortsman et al., 2022; Jin et al., 2022). This approach typically assumes uniform network architectures and attempts to establish mappings between the weights of distinct neural net-

---

*   Work was done during the internship at Tencent AI lab.
†   Corresponding authors.

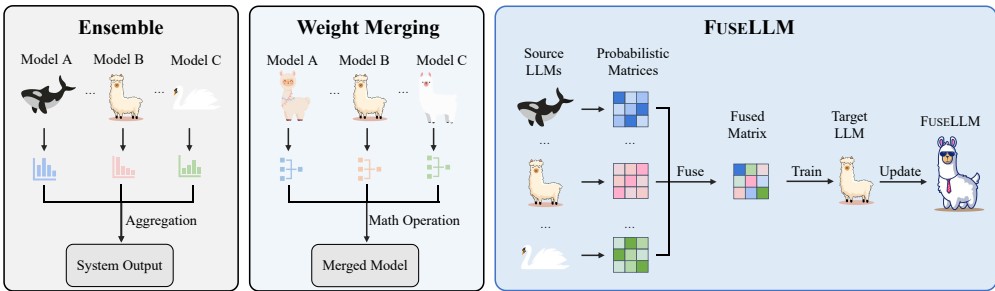

Figure 1: Illustration of conventional model fusion techniques (ensemble and weight merging) and our knowledge fusion approach for LLMs (FUSELLM). Different animal icons represent different LLMs, with various species denoting LLMs possessing differing architectures. FUSELLM externalizes the knowledge from multiple LLMs and transfers their capabilities to a target LLM.

works, which is often unattainable in the context of LLMs. Moreover, weight merging may lead to suboptimal results when substantial differences exist in the parameter space (Li et al., 2022).

In this paper, we explore the fusion of LLMs from a probabilistic distribution perspective. For an input text, we argue that the probabilistic distributions generated by different source LLMs can reflect their inherent knowledge in understanding this text. Therefore, the proposed FUSELLM leverages the generative distributions of source LLMs to externalize both their collective knowledge and individual strengths and transfer them to the target LLM through lightweight continual training. To achieve this, we develop a new strategy for aligning tokenizations originating from different LLMs and explore two methods for fusing the probability distributions generated by these diverse LLMs. During the continual training, FUSELLM places significant emphasis on minimizing the divergence between the target LLM's probabilistic distributions and those of the source LLMs.

To empirically demonstrate the effectiveness of FUSELLM, we examine a challenging yet general scenario of LLMs fusion, where the source models share minimal commonalities. Specifically, we focus on three popular open-source LLMs that possess distinct architectures and functionalities: Llama-2 (Touvron et al., 2023), OpenLLaMA (Geng & Liu, 2023), and MPT (Team, 2023). Evaluations across three benchmarks, which consist of a total of 42 tasks spanning reasoning, commonsense, and code generation, confirm that the target model trained by our method outperforms each source LLM and the baseline in most tasks. Moreover, we simulate the existence of functionally distinct LLMs with identical architecture by continually training a single base model on several domain-specific corpora. When evaluated based on perplexity, our method demonstrates superior potential in combining the capabilities of these structurally identical LLMs compared to traditional ensemble and weight merging methods.

To sum up, this paper explores a novel challenge called LLMs fusion, with the goal of creating a unified model that effectively utilizes the collective capabilities and unique strengths of diverse LLMs. Illustrated in Figure 1, our proposed approach distinguishes itself from traditional ensemble and weight merging techniques by prioritizing the fusion of multiple LLMs through knowledge externalization and transfer. This study yields several findings that may spark future research. Firstly, while we demonstrate the effectiveness of our method through lightweight continual training on a compact, high-quality corpus, the thoughtful selection of the training corpus can be a crucial consideration, particularly with regard to its relevance to downstream tasks. Secondly, in scenarios where the capabilities of source LLMs vary significantly, the fusion function appears to be crucial in effectively combining their respective strengths. Lastly, when compared to traditional model ensemble and merging techniques, the field of LLMs fusion appears to be a more promising avenue for exploration, especially in light of the diverse structures and substantial model sizes of LLMs.

## 2  RELATED WORK

**Model Fusion**  The integration of capabilities from diverse models has been a long-standing objective, with existing approaches mainly falling into two categories. Firstly, the traditional technique of model *ensemble* combines the outputs of multiple models to enhance overall system performance (Littlestone & Warmuth, 1994; Sagi & Rokach, 2018). Note that this technique doesn't involve the explicit merging of multiple models into a new one. Common methods for model ensemble

typically employ weighted averaging (Littlestone & Warmuth, 1994) or majority voting (Monteith et al., 2011) to consolidate predictions from various models. Recently, Jiang et al. (2023) introduced an ensemble framework designed to leverage the diverse strengths of multiple open-source LLMs. This framework first employs a pairwise comparison method to detect subtle distinctions among candidate outputs. Then, it combines the top-ranked candidates to produce an enhanced output, capitalizing on their strengths while mitigating their weaknesses.

Secondly, *weight merging* presents another approach that facilitates model fusion at the parameter level. Gupta et al. (2020) and Wortsman et al. (2022) merged weights from models with identical structures, obtained through different strategies or configurations, to achieve improved overall performance. Similarly, Cha et al. (2021), Rame et al. (2022), and Arpit et al. (2022) explored weighted averaging of models derived from different configurations to enhance out-of-distribution generalization. Furthermore, Jin et al. (2022) merged models designed for specific domains or tasks to create a generalist capable of addressing all domains or tasks. Going beyond parameter merging of entire models, Wang et al. (2022b), Huang et al. (2023), and Zhang et al. (2023) applied linear mathematical operations to adapter parameters to achieve superior generalization performance.

In a nutshell, while model ensemble requires the parallel deployment of multiple models, weight merging is generally limited to models with identical architectures. In contrast, the approach proposed in this paper supports the fusion of multiple LLMs with diverse architectures by explicitly transferring their knowledge and capabilities to a target LLM.

**Knowledge Distillation**  Knowledge distillation (Hinton et al., 2015), initially proposed for model compression, involves training a student model under the guidance of one or more teacher models. In the NLP community, knowledge distillation has been widely applied to text classification tasks. These applications include training the student model to replicate the teacher's output distribution (Sanh et al., 2019; Turc et al., 2019), as well as features (Sun et al., 2019; Jiao et al., 2020) and relations (Wang et al., 2020) derived from intermediate layers of the teacher model. In the realm of text generation, the conventional approach focuses on minimizing the KL divergence between the student and teacher generation distributions. This is achieved by using the teacher's probability distributions at each time step as supervision (Khanuja et al., 2021; Gu et al., 2023; Agarwal et al., 2023) or by directly training on the teacher's generated texts (Peng et al., 2023; Xu et al., 2023).

While our method shares a framework similar to multi-teacher knowledge distillation, there are two significant distinctions. First, in traditional knowledge distillation, the student models are typically constrained to be smaller in size than the teachers. In our scenario, however, there are no limitations on the size of the target model. Second, traditional knowledge distillation often results in the student models lagging behind the teachers in performance after distillation. In contrast, we anticipate that after the fusion, the target model will surpass any of the source models in performance.

## 3 KNOWLEDGE FUSION OF LLMs

The primary objective of LLMs fusion is to externalize the collective knowledge embedded within multiple source LLMs and integrate their capabilities into a target LLM. Given $K$ source LLMs $\{\mathcal{M}_j^s\}_{j=1}^K$ with varying architectures, each having undergone individual pre-training or fine-tuning on distinct datasets, the key idea behind our approach is to initially stimulate LLMs to manifest their inherent knowledge by challenging them to predict the next token. The probabilistic distributions of these predictions are thoroughly assessed, and the most accurate predictions are utilized to continually train the target LLM $\mathcal{M}^t$ on a corpus $\mathcal{C}$ using the causal language modeling objective. In the following sections, we start with a brief introduction to the preliminaries, followed by a detailed explanation of our LLMs fusion framework. Finally, we delve into the implementation details.

### 3.1 PRELIMINARIES

Let $t$ denote a text sequence of length $N$ sampled from the corpus $\mathcal{C}$ and $t_{<i} = (t_1, t_2, \ldots, t_{i-1})$ denote the sequence preceding the $i$th token. The causal language modeling (CLM) objective for training a language model parameterized by $\theta$ is defined as minimizing the negative log-likelihood:

$$\mathcal{L}_{\text{CLM}} = -\mathbb{E}_{t \sim \mathcal{C}} \left[ \sum_i \log p_\theta(t_i | t_{<i}) \right], \tag{1}$$

where $p_\theta(t_i|t_{<i})$ is the model's predicted probability for token $t_i$ given the preceding tokens.

The above objective decomposes sequence likelihood into token-level cross-entropy losses, comparing each token's predicted distribution to its one-hot representation. To provide a more generalized perspective, we reframe this token-level view into a sequential distribution format. Specifically, for the text sequence $t$, we aggregate token-level predictions and create a probabilistic distribution matrix, $\mathbf{P}_t^\theta \in \mathbb{R}^{N \times V}$, where the $i$-th row represents the distribution predicted by the model for the $i$th token over the vocabulary of size $V$. The CLM objective can then be interpreted as reducing the discrepancy between $\mathbf{P}_t^\theta$ and the one-hot label matrix, $\mathbf{O}_t \in \{0, 1\}^{N \times V}$, where each row is a one-hot representation of the corresponding gold token. Formally, the CLM objective is transformed into the following representation:

$$\mathcal{L}_{\text{CLM}} = -\mathbb{E}_{t \sim \mathcal{C}} \left[ \mathbb{D}(\mathbf{P}_t^\theta, \mathbf{O}_t) \right], \tag{2}$$

where $\mathbb{D}(\cdot, \cdot)$ represents the discrepancy function between two matrices, and it is equivalent to Eq. 1 when implemented as the KL divergence.

## 3.2 LLMs Fusion

Taking this perspective on a language model, we argue that the probabilistic distribution matrix can reflect its certain inherent knowledge in understanding the text. Consequently, different probabilistic distribution matrices for the same text, originating from various LLMs, can be used to represent the diverse knowledge embedded within these models. Acknowledging this, the proposed FuseLLM approach tackles LLMs fusion through probabilistic modeling, aiming to create a unified LLM by merging the probabilistic distributions of the source LLMs. To achieve this, when starting with a set of LLMs to fuse, FuseLLM undergoes lightweight continual training of the target LLM on a raw text corpus that mirrors the pre-training dataset. Instead of relying solely on the CLM objective, FuseLLM places significant emphasis on minimizing the divergence between the target LLM's probabilistic distributions and those of the source LLMs.

For each text in the corpus $\mathcal{C}$, we apply the provided $K$ source LLMs and obtain a set of probabilistic distribution matrices, denoted as $\{\mathbf{P}_t^{\theta_j}\}_{j=1}^K$, where $\theta_j$ represents the parameters of the $j$th LLM. Utilizing these matrices, we externalize the knowledge from individual models into a unified space, essentially creating unified probabilistic representations over the text. We acknowledge that variances in vocabulary among the source LLMs can lead to misaligned matrices $\{\mathbf{P}_t^{\theta_j}\}_{j=1}^K$. To address this, we employ a token alignment strategy, which is explained in Section 3.3, to foster more coherent probabilistic interpretations across models.

Having aligned the probabilistic matrices, we proceed to fuse them into a single compact representation. Various fusion strategies can be applied for this purpose, as detailed in Section 3.3. We use $\mathbf{P}_t$ to represent the fused representation matrix as follows:

$$\mathbf{P}_t = \mathbb{F}\text{usion}(\mathbf{P}_t^{\theta_1}, \mathbf{P}_t^{\theta_2}, \dots, \mathbf{P}_t^{\theta_K}), \tag{3}$$

where $\mathbb{F}\text{usion}(\cdot)$ denotes the function that combines multiple matrices, and the resulting matrix $\mathbf{P}_t$ is seen as a representation of the collective knowledge and distinctive strengths of the source LLMs.

To transfer the capabilities of source LLMs to the target LLM, we enforce alignment between the target LLM's predictions and the fused representation matrix $\mathbf{P}_t$. We use $\mathbf{Q}_t$ to represent the output distribution matrix of the target LLM for text $t$, and then define our fusion objective as follows:

$$\mathcal{L}_{\text{Fusion}} = -\mathbb{E}_{t \sim \mathcal{C}} \left[ \mathbb{D}(\mathbf{Q}_t, \mathbf{P}_t) \right]. \tag{4}$$

The overall objective for our continual training consists of a weighted combination of the causal language modeling objective $\mathcal{L}_{\text{CLM}}$ and the fusion objective $\mathcal{L}_{\text{Fusion}}$ as follows:

$$\mathcal{L} = \lambda \mathcal{L}_{\text{CLM}} + (1 - \lambda) \mathcal{L}_{\text{Fusion}}. \tag{5}$$

## 3.3 Implementation of FuseLLM

In this section, we present the implementation details of token alignment and the fusion function for fusing different LLMs in our FuseLLM method.

**Token Alignment**  Ensuring token alignment across multiple LLMs is crucial for effective knowledge fusion, as it guarantees proper mapping of probabilistic distribution matrices. Fu et al. (2023) employed dynamic programming to recursively minimize the total cost of editing one sequence of tokens to match the other. If a one-to-one mapping exists between two tokens, the corresponding distributions are perfectly mapped. Otherwise, the mapped distribution degenerates into a one-hot vector. Since tokens generated by different tokenizers for the same sequence typically exhibit limited differences, we propose to enhance the success rate of token alignment by replacing the exact match (EM) constraint in Fu et al. (2023) with a minimum edit distance (MinED) strategy, which maps tokens from different tokenizers based on MinED. This relaxation of token alignment helps preserve substantial information in the distribution matrices while introducing minor errors. For more details of the token alignment, please refer to Appendix A.

**Fusion Strategies**  To combine the collective knowledge of source LLMs while preserving their unique strengths, it is essential to evaluate the quality of different LLMs and assign varying levels of importance to their respective distribution matrices. For this purpose, when dealing with text $t$, we utilize the cross-entropy loss between the distribution matrices and the gold labels as an indicator of the prediction quality of the LLMs (Marion et al., 2023). A lower cross-entropy score for a source LLM signifies a more accurate understanding of the text, and its prediction should be accorded greater significance. Based on this criterion, we introduce two fusion functions: (1) MinCE: This function outputs the distribution matrix with the minimum cross-entropy score; (2) AvgCE: This function produces a weighted average of the distribution matrices based on cross-entropy scores.

The complete process of the FUSELLM method is described in Algorithm 1.

---

**Algorithm 1** FUSELLM for LLMs Fusion

---

**Require:** Source LLMs $\{\mathcal{M}_j^s\}_{j=1}^K$, training corpus $\mathcal{C}$.
 1: Initialize the target LLM $\mathcal{M}^t$ with one of the source LLMs.
 2: **for** text $t$ in $\mathcal{C}$ **do**
 3:     Apply the $K$ source LLMs to compute probabilistic distribution matrices $\{\mathbf{P}_t^{\theta_j}\}_{j=1}^K$.
 4:     Align $\{\mathbf{P}_t^{\theta_j}\}_{j=1}^K$ using the MinED alignment method.
 5:     Fuse $\{\mathbf{P}_t^{\theta_j}\}_{j=1}^K$ to obtain $\mathbf{P}_t$ with the MinCE or AvgCE fusion function.
 6:     Update parameters of $\mathcal{M}^t$ by minimizing the overall loss fuction in Eq. 5.
 7: **end for**
 8: **return** $\mathcal{M}^t$.

---

## 4  EXPERIMENTS

In our experiments, we consider a general but challenging scenario of LLMs fusion where the source models share minimal commonalities in architectures or functionalities. Specifically, we conduct experiments on the 7B scale and select three representative open-source models: Llama-2, Open-LLaMA, and MPT as the source LLMs for fusion. Regarding the target LLM, we opt for another Llama-2 7B, which is generally the most robust one among the three source LLMs. The target LLM starts with the same pre-trained weights as its source counterpart but differs in that it updates parameters during training. To evaluate the performance of FUSELLM, we conduct experiments on benchmarks assessing the capabilities of LLMs in reasoning, commonsense, and code generation.

### 4.1  EXPERIMENTAL SETUP

**Dataset for continual training**  To continually train the target LLM for LLMs fusion, it is essential to have a compact yet diverse training dataset. We have chosen MiniPile, a meticulously curated dataset resulting from a thorough clustering and filtering process. MiniPile comprises approximately 1 million documents across 22 domains and 1.8 billion tokens, constituting less than 0.1% of the 2 trillion training tokens of Llama-2. More dataset details can be found in Appendix B.

**Fusion function**  For the fusion function, we use the minimum cross-entropy (MinCE). However, the impact of employing alternative fusion functions will be examined in Section 4.4.

**Training details**  We train the target LLM of Llama-2 7B using a batch size of 128 and a maximum length of 2048 on a single node equipped with 8 NVIDIA A100 GPUs, each with 40GB of memory.

Our training framework is implemented based on the Huggingface Transformers (Wolf et al., 2020) and accelerated with FlashAttention (Dao et al., 2022). We empirically set the combination weight $\lambda$ in Eq. 5 to 0.9. The training consists of only a single epoch, which takes approximately 33 hours. For further hyper-parameter details, please refer to Appendix C.

**Evaluation**    We evaluate FUSELLM on three benchmarks that represent different core capabilities of LLMs, spanning *reasoning*, *commonsense*, and *code generation*.

• Big-Bench Hard (BBH) (Suzgun et al., 2022) is a benchmark to evaluate the general *reasoning* ability of LLMs. It contains 23 multiple-choice tasks and 4 free-form generation tasks from the Big-Bench (Srivastava et al., 2022), which can be classified into four categories: algorithmic and arithmetic reasoning, natural language understanding, world knowledge, and multilingual knowledge and reasoning. We follow previous work (Wang et al., 2023b) to generate the predictions based on few-shot chain-of-thought (CoT) prompts and then calculate the exact match (EM) accuracy.

• Common Sense (CS) is a benchmark to evaluate the *commonsense* capability of LLMs. We consider 5 standard multiple-choice tasks: ARC easy and challenge (Clark et al., 2018), BoolQ (Clark et al., 2019a), HellaSwag (Zellers et al., 2019), and OpenBookQA (Mihaylov et al., 2018). We employ lm-eval-hardness (Gao et al., 2021) to conduct a likelihood-based zero-shot evaluation. Specifically, we select the option with the highest likelihood given the context and report the accuracy.

• MultiPL-E (ME) (Cassano et al., 2022) is a multilingual programming benchmark to assess the *coding* ability of LLMs. It is translated from the Python benchmark (Chen et al., 2021) into parallel datasets in 18 programming languages. We use the bigcode-evaluation-hardness (Ben Allal et al., 2022) to perform zero-shot code generation in 10 popular programming languages in the HumanEval category and report the pass@1 (Chen et al., 2021) based on 20 generated samples for each question.

**Baselines**    In our experiments, we compare our FUSELLM with two sets of baselines: (1) **original LLMs**, including Llama-2 7B, OpenLLaMA 7B, and MPT 7B; and (2) **Llama-2 CLM**: continually trained Llama-2 7B on MiniPile using only the casual language modeling objective.

## 4.2    OVERALL RESULTS

Table 1 presents the overall results of FUSELLM in comparison to the baseline methods on BBH. We can observe that the three source LLMs exhibit varying performance across the 27 BBH tasks, with Llama-2 generally outperforming the others. After continual training with a compact and diverse corpus, Llama-2 CLM shows a relative improvement of 1.86% compared to Llama-2, although this improvement is relatively modest and inconsistent across tasks. On average, FUSELLM demonstrates an average relative performance gain of 5.16% over the original Llama-2 across all 27 tasks. In specific tasks, the enhancements achieved by FUSELLM are substantial (e.g., from 54.40 to 65.20 in the Hyperbaton task). In tasks such as Dick Languages where simple continual pre-training leads to a decline in performance, FUSELLM leverages the combined strengths of individual source LLMs to recover performance improvements. Note that FUSELLM occasionally exhibits degraded performance on tasks such as Geometric Shapes and Word Sorting, which could be attributed to two reasons. First, the other source LLMs, apart from Llama-2, perform poorly on these tasks, affecting the fusion results. Second, the relevance between the continual training dataset and downstream tasks also contributes to the performance degradation.

Table 2 shows the zero-shot performance of FUSELLM and the baseline methods on the Common Sense (CS) benchmark. The results demonstrate that FUSELLM consistently surpasses the baselines across all five tasks, achieving a relative performance improvement of 1.25% over Llama-2. In contrast, Llama-2 CLM exhibits a marginal improvement, with only a 0.16% relative enhancement compared to Llama-2. Notably, substantial improvements from Llama-2 to FUSELLM are observed in the challenging ARC-challenge (2.40%) and OpenBookQA (2.71%) tasks, highlighting the effectiveness of FUSELLM in leveraging collective knowledge to address intricate problems.

For the code generation evaluation, the zero-shot performance of FUSELLM on the MultiPL-E (ME) benchmark is reported in Table 3. We observe that FUSELLM outperforms Llama-2 in 9 out of the 10 tasks, with a notable enhancement in the pass@1 score for specific programming languages such as R, increasing from 4.97 to 5.84. Given that both OpenLLaMA and MPT demonstrate remarkable performances in code generation tasks compared to Llama-2, the fusion result via FUSELLM achieves an average performance gain of 6.36%, which is considerably higher than the 1.37% improvement observed in Llama-2 CLM. However, it's important to note that FUSELLM still exhibits

| Task | OpenLLaMA | MPT | Llama-2 | Llama-2 CLM | FUSELLM |
|---|---|---|---|---|---|
| Boolean Expressions | 74.40 | 66.00 | 68.80 | **76.00** (+10.47%) | 71.60 (+4.07%) |
| Causal Judgement | 45.45 | **50.80** | **50.80** | 46.52 (-8.43%) | 46.52 (-8.43%) |
| Date Understanding | 43.60 | 43.60 | 59.60 | 59.20 (-0.67%) | **62.40** (+4.70%) |
| Disambiguation QA | 36.00 | 47.60 | 46.80 | 48.00 (+2.56%) | **50.00** (+6.84%) |
| Dyck Languages | 5.20 | 5.20 | 7.20 | 6.40 (-11.11%) | **8.80** (+22.22%) |
| Formal Fallacies | 50.80 | **52.80** | 49.20 | 48.80 (-0.81%) | 49.20 (+0.00%) |
| Geometric Shapes | 0.00 | 0.00 | **34.40** | 19.20 (-44.17%) | 22.80 (-33.72%) |
| Hyperbaton | 62.80 | 53.60 | 54.40 | 56.40 (+3.68%) | **65.20** (+19.85%) |
| Logical Deduction (3 objects) | 43.60 | 40.80 | 54.00 | 57.20 (+5.93%) | **60.40** (+11.85%) |
| Logical Deduction (5 objects) | 24.80 | 31.60 | 31.20 | **35.60** (+14.10%) | 33.20 (+6.41%) |
| Logical Deduction (7 objects) | 16.80 | 18.40 | 24.80 | **29.60** (+19.35%) | 25.60 (+3.23%) |
| Movie Recommendation | 39.60 | 52.00 | 72.80 | 71.60 (-1.65%) | **73.60** (+1.10%) |
| Multistep Arithmetic Two | 0.80 | 0.40 | 0.80 | 4.40 (+450.00%) | **4.80** (+500.00%) |
| Navigate | 54.00 | 48.80 | 56.00 | 61.20 (+9.29%) | **64.40** (+15.00%) |
| Object Counting | 49.60 | 40.40 | 49.60 | 51.60 (+4.03%) | **55.20** (+11.29%) |
| Penguins in a Table | 28.08 | 28.08 | 32.19 | 31.51 (-2.11%) | **32.88** (+2.14%) |
| Reasoning about Colored Objects | 28.00 | 31.60 | 46.40 | 47.20 (+1.72%) | **48.40** (+4.31%) |
| Ruin Names | 31.20 | 23.20 | **34.00** | 30.80 (-9.41%) | 32.40 (-4.71%) |
| Salient Translation Error Detection | 14.80 | 0.00 | 24.80 | 27.60 (+11.29%) | **29.20** (+17.74%) |
| Snarks | 44.94 | 45.51 | 47.75 | **49.44** (+3.54%) | **49.44** (+3.54%) |
| Sports Understanding | 64.40 | 82.40 | 90.00 | 90.00 (+0.00%) | **91.20** (+1.33%) |
| Temporal Sequences | **32.00** | 21.20 | 12.80 | 16.40 (+28.13%) | 16.40 (+28.13%) |
| Tracking Shuffled Objects (3 objects) | **36.40** | 30.40 | 33.20 | 33.20 (+3.61%) | 34.40 (+3.61%) |
| Tracking Shuffled Objects (5 objects) | **19.20** | 14.40 | 15.60 | 15.20 (-2.56%) | 15.60 (+0.00%) |
| Tracking Shuffled Objects (7 objects) | 10.80 | 2.00 | **11.20** | 9.60 (-14.29%) | 10.40 (-7.14%) |
| Web of Lies | 51.60 | 63.60 | 50.80 | 61.60 (+21.26%) | **65.60** (+29.13%) |
| Word Sorting | 5.60 | 6.80 | **12.80** | 7.60 (-40.63%) | 7.60 (-40.63%) |
| Avg. 27 Tasks | 33.87 | 33.38 | 39.70 | 40.44 (+1.86%) | **41.75** (+5.16%) |

Table 1: Overall results of FUSELLM and baselines in reasoning evaluations on Big-Bench Hard (BBH), where percentages indicate the rate of improvement/decrease compared to Llama-2.

| Task | OpenLLaMA | MPT | Llama-2 | Llama-2 CLM | FUSELLM |
|---|---|---|---|---|---|
| ARC-easy | 69.70 | 70.12 | 74.58 | 74.54 (-0.05%) | **75.04** (+0.62%) |
| ARC-challenge | 41.38 | 42.15 | 46.33 | 46.50 (+0.37%) | **47.44** (+2.40%) |
| BoolQ | 72.29 | 74.74 | 77.71 | 76.88 (-1.07%) | **78.13** (+0.54%) |
| HellaSwag | 74.53 | 76.25 | 76.00 | 76.57 (+0.75%) | **76.78** (+1.03%) |
| OpenBookQA | 41.00 | 42.40 | 44.20 | 44.80 (+1.36%) | **45.40** (+2.71%) |
| Avg. 5 Tasks | 59.78 | 61.13 | 63.76 | 63.86 (+0.16%) | **64.56** (+1.25%) |

Table 2: Overall results of FUSELLM and baselines in commonsense evaluations on CommenSense (CS), where percentages indicate the rate of improvement/decrease compared to Llama-2.

| Task | OpenLLaMA | MPT | Llama-2 | Llama-2 CLM | FUSELLM |
|---|---|---|---|---|---|
| C++ | **14.47** | 13.11 | 7.45 | 9.88 (+32.62%) | 9.25 (+24.16%) |
| Go | **68.20** | 66.96 | 57.02 | 54.44 (-4.52%) | 59.78 (+4.84%) |
| Java | **14.28** | 13.42 | 10.31 | 10.50 (+1.84%) | 10.34 (+0.29%) |
| JavaScript | **17.61** | 13.01 | 13.17 | 14.25 (+8.20%) | 14.32 (+8.73%) |
| PHP | **11.24** | 9.53 | 9.75 | 9.04 (-7.28%) | 9.41 (-3.49%) |
| Python | 15.96 | **17.24** | 13.85 | 13.07 (-5.63%) | 13.91 (+0.43%) |
| R | **7.52** | 4.53 | 4.97 | 5.25 (+5.63%) | 5.84 (+17.51%) |
| Ruby | 10.34 | **12.33** | 10.37 | 10.68 (+2.99%) | 11.24 (+8.39%) |
| Rust | 6.18 | **8.29** | 6.77 | 6.96 (+2.81%) | 7.05 (+4.14%) |
| TypeScript | **15.31** | 14.13 | 12.61 | 14.19 (+12.53%) | 14.50 (+14.99%) |
| Avg. 10 Tasks | **18.11** | 17.26 | 14.63 | 14.83 (+1.37%) | 15.56 (+6.36%) |

Table 3: Overall results of FUSELLM and baselines in code generation evaluations on MultiPL-E (ME), where percentages indicate the rate of improvement/decrease compared to Llama-2.

a performance gap compared to OpenLLaMA or MPT in this evaluation. This discrepancy can be attributed to two primary reasons: the inferior performances of Llama-2 as the target model compared to other source LLMs and an insufficient proportion of coding-related texts in the continual training corpus, estimated at approximately 7.59%[1].

---

[1] Since MiniPile lacks specific data percentages for individual domains, we approximate this by considering the percentage of the Github domain in The Pile.

## 4.3 THE FUSED PROBABILISTIC DISTRIBUTIONS

We investigate the effectiveness of the fused probabilistic distributions obtained from multiple LLMs and track the trend of performance improvement during the training process. Figure 2 illustrates the comparison of few-shot CoT performance between Llama-2 CLM and FUSELLM with varying scales of training data on BBH. Our observations reveal that FUSELLM enhances the exact match (EM) accuracy by 2.5% compared to Llama-2 CLM and achieves the best performance of Llama-2 CLM within 0.52 billion tokens. Notably, this represents a $3.9\times$ reduction in token requirements compared to the 1.57 billion tokens needed by Llama-2 CLM. These results suggest that the probabilistic distributions derived from LLMs contain knowledge that is more readily learnable than the original text sequences, which accelerates the optimization process.

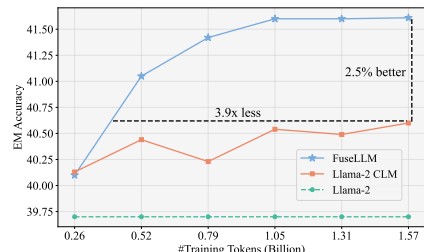

Figure 2: Effect of the fused distributions in accelerating the optimization process on BBH, where the x-axis denotes the number of training tokens and the y-axis denotes the exact match accuracy.

This finding aligns with the observations in Hsieh et al. (2023). We further conduct an experiment to show that our performance improvement stems from the integration of knowledge from multiple LLMs rather than solely from continual training. The results and analysis are shown in Appendix G.

## 4.4 ANALYSIS OF IMPLEMENTATION PROCESS

In this section, we delve into the crucial elements of FUSELLM's implementation, including the number of source LLMs, the criteria for token alignment, and the choice of the fusion function.

**Number of source LLMs.** In Table 4, we present the results of fusing different numbers of LLMs. We note that the performance of FUSELLM demonstrates apparent improvement as the number of models increases from 1 to 3. Nevertheless, the benefits of integrating additional models exhibit variations across benchmarks. Remarkably, a consistent performance improvement is observed in BBH.

| Model | BBH | CS | ME |
|---|---|---|---|
| OpenLLaMA | 33.87 | 59.78 | 18.11 |
| MPT | 33.38 | 61.13 | 17.26 |
| Llama-2 | 39.70 | 63.76 | 14.63 |
| Llama-2 CLM | 40.44 (+1.86%) | 63.86 (+0.16%) | 14.83 (+1.37%) |
| Llama-2 + OpenLLaMA | 41.00 (+3.27%) | 64.50 (+1.16%) | 15.51 (+6.02%) |
| Llama-2 + MPT | 41.16 (+3.68%) | 64.51 (+1.18%) | 15.47 (+5.74%) |
| FUSELLM | 41.75 (+5.16%) | 64.56 (+1.25%) | 15.56 (+6.36%) |

Table 4: Results of FUSELLM by incorporating varying numbers of models.

Whereas in CS or ME, the advantages are more prominent when fusing two models. This phenomenon may be attributed to the considerable performance differences among the three models on various tasks in BBH, while the performance differences in tasks of CS or ME are relatively smaller.

**Criteria for token alignment.** During the fusion of LLMs, ensuring the proper alignment of tokens and vocabularies from multiple models is of paramount importance. In Table 5 (upper), we present a comparison of two alignment criteria. It is evident that the proposed MinED method, which is based on minimum edit distance, consistently outperforms the EM method introduced by Fu et al. (2023), which relies on exact matching. We suggest that this performance enhancement results from MinED's ability to relax the constraints of EM, as tokens separated by distinct tokenizers within the same sequence often exhibit minor discrepancies. Consequently, MinED effectively supplements a considerable amount of useful token information while introducing negligible errors.

**Fusion function.** In Section 3.3, we introduce two variations of the fusion function for FUSELLM: one utilizing a distribution matrix with minimum cross entropy (MinCE) and the other adopting a weighted average of distribution matrices based on cross entropy (AvgCE). A comparison of the two functions is presented in Table 5 (down). The findings demonstrate that FUSELLM with MinCE consistently outperforms AvgCE across all benchmarks. This

| Choice | BBH | ME | CS |
|---|---|---|---|
| | *Alignment Criteria* | | |
| EM | 41.57 | 15.49 | 64.24 |
| MinED | 41.75 (+0.43%) | 15.56 (+0.45%) | 64.56 (+0.50%) |
| | *Fusion Function* | | |
| AvgCE | 41.04 | 15.39 | 63.98 |
| MinCE | 41.75 (+1.73%) | 15.56 (+1.10%) | 64.56 (+0.91%) |

Table 5: Comparison of different token alignment criteria (upper) and fusion functions (down).

can be attributed to the distortions introduced by the straightforward weighted summation used in AvgCE, which may diminish the distinct advantages of individual LLMs.

### 4.5 FUSELLM VS. KNOWLEDGE DISTILLATION

While knowledge distillation techniques can also be utilized to enhance a LLM's capabilities, FUSELLM stands out due to two distinct aspects, as previously outlined. In this section, we compare FUSELLM with traditional knowledge distillation. Specifically, we extract probabilistic distributions from Llama-2 13B and apply the conventional knowledge distillation method to transfer its abilities into Llama-2 7B. As illustrated in Table 6, the distilled model (Llama-2 KD) outperforms the original Llama-2 7B across all benchmarks, demonstrating the effectiveness of knowledge distillation. How-

| Model | BBH | CS | ME |
|---|---|---|---|
| Llama-2 13B | 47.92 | 66.33 | 18.76 |
| OpenLLaMA | 33.87 | 59.78 | 18.11 |
| MPT | 33.38 | 61.13 | 17.26 |
| Llama-2 | 39.70 | 63.76 | 14.63 |
| Llama-2 CLM | 40.44 (+1.86%) | 63.86 (+0.16%) | 14.83 (+1.37%) |
| Llama-2 KD | 40.88 (+2.97%) | 64.41 (+1.02%) | 15.45 (+5.60%) |
| FUSELLM | 41.75 (+5.16%) | 64.56 (+1.25%) | 15.56 (+6.36%) |

Table 6: Comparison of FUSELLM and knowledge distillation. Llama-2 KD denotes the enhanced Llama-2 7B achieved via knowledge distillation from Llama-2 13B. Percentages indicate the rate of improvement compared to Llama-2.

ever, when compared to FUSELLM, the improvement achieved by Llama-2 KD is relatively modest, especially in the case of BBH (2.97% vs. 5.16%). This suggests that the superior results achieved by FUSELLM through the integration of three 7B models with diverse architectures via continual training outweigh the benefits of simply distilling knowledge from a single 13B model. This observation highlights the idea that "More is different, but different can also be more" (Tay et al., 2022).

### 4.6 FUSELLM VS. ENSEMBLE/MERGING

As previously mentioned, conventional techniques such as model ensemble and weight merging are commonly employed to fuse multiple LLMs. To compare the efficacy of our FUSELLM with these existing fusion methods, we conduct experiments simulating scenarios where multiple LLMs originated from the same base model but were trained on distinct corpora. We first select three relevant domains (PhilPapers, NIH ExPorter, and USPTO Backgrounds) from The Pile and use 1 billion tokens from

| Model | Phil | NIH | USPTO | Average |
|---|---|---|---|---|
| Pythia | 0.9008 | 0.6740 | 0.6077 | 0.7275 |
| Phil | **0.8397** | 0.6861 | 0.6228 | 0.7162 |
| NIH | 0.9248 | **0.6215** | 0.6278 | 0.7247 |
| USPTO | 0.9296 | 0.6872 | **0.6017** | 0.7395 |
| Ensemble | 0.8960 | 0.6647 | 0.6180 | 0.7262 |
| Weight Merging | 0.8786 | 0.6496 | 0.6054 | 0.7112 |
| FUSELLM | 0.8463 | 0.6569 | 0.6068 | **0.7034** |

Table 7: Comparison of perplexity between FUSELLM and ensemble&weight merging.

each domain to continually train Pythia 1B (Biderman et al., 2023), resulting in three distinct LLMs with identical structures. Then, we apply different fusion techniques to these LLMs: (1) The ensemble method calculates a weighted average of the probabilities generated by all LLMs, considering the performance of each model; (2) The weight merging method merges multiple LLMs into a single one within the parameter space, with the merging weights determined by model performance; (3) FUSELLM undergoes continual training on 0.1 billion tokens sampled from the three domains. The results of perplexity for FUSELLM and the other fusion methods on the test sets are presented in Table 7. We measure perplexity in bits per UTF-8 encoded byte (BPB) following the implementation in The Pile. We observe that after training with 1 billion tokens, the capabilities of the original LLM are transferred to each domain-specific LLM, resulting in decreased performance in other domains. While all fusion techniques can integrate the strengths of diverse models, FUSELLM consistently achieves the lowest average perplexity across the three domains. This underscores its potential for harnessing collective knowledge more effectively than ensemble and weight merging methods.

## 5 CONCLUSION

In this study, we have explored the realm of knowledge fusion for LLMs to create a unified model that combines the capabilities and distinctive strengths of multiple structurally diverse LLMs. We introduced a novel method, FUSELLM, which leverages the generative distributions of these source LLMs to externalize their knowledge and employs them in the continual training of the target LLM. Through a series of experiments, we have demonstrated the superiority of FUSELLM over individual source LLMs and established baselines. Notably, in a simulated experiment featuring multiple structurally identical LLMs, FUSELLM has showcased its competitive effectiveness compared to ensemble and weight merging methods. Hence, the domain of LLMs fusion emerges as a more promising avenue for exploration, particularly given the diverse structures and substantial model sizes of LLMs. We believe that these findings will inspire future research endeavors.

ACKNOWLEDGEMENTS

This work was supported by the National Natural Science Foundation of China (No. 62176270), the Guangdong Basic and Applied Basic Research Foundation (No. 2023A1515012832), and the Tencent AI Lab Rhino-Bird Focused Research Program.

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

## A DETAILS OF TOKEN ALIGNMENT

For an input text, token alignment involves aligning two distribution matrices from two source LLMs. Therefore, the alignment comprises two dimensions: token-wise with respect to the text and distribution-wise with respect to the vocabulary. To provide a clear explanation, we show an example of different methods for token alignment in Figure 3.

In the token dimension, we utilize the dynamic programming approach to recursively minimize the total cost of editing one sequence of tokens to align with another. When the mapped tokens are identical, such as the token "now" in the given example, these tokens are successfully aligned, allowing for the corresponding distributions to align subsequently. However, when the mapped tokens exhibit differences, such as the "get" and "gets" tokens in the example, the previous EM method proposed by Fu et al. (2023) does not align these tokens, resulting in the distributions degenerating into one-hot vectors. In contrast, our proposed MinED method successfully aligns the "gets" token with the "get" token, as they exhibit the minimal edit distance in the vocabularies from the two source LLMs.

Concerning the distribution dimension, the alignment is performed between two vocabularies from different tokenizers of two source LLMs. Therefore, for distribution values with identical tokens, such as "current 0.05" and "current 0.04", they will be aligned effectively. For distribution values involving different tokens, such as "immediate 0.04" and "immediately 0.03", the EM method disregards this value. However, the proposed MinED method maps "immediately" to "immediate" due to their minimal edit distance, resulting in the successful alignment of these distribution values.

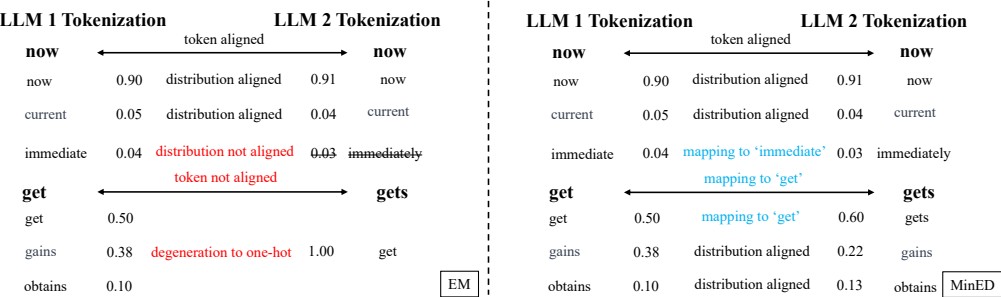

Figure 3: An example of different methods for token alignment.

## B DETAILS OF MINIPILE

MiniPile is curated from The Pile (Gao et al., 2020) through a three-stage pruning process: (1) extracting embeddings for all documents with E5-Large (Wang et al., 2022a), which is a sentence embedding model, (2) clustering the embeddings using K-means, and (3) filtering out low-quality clusters. Therefore, MiniPile retains a compact scale while exhibiting extensive diversity, making it a prevalent choice for efficient training of LLMs (Kaddour et al., 2023; Sanyal et al., 2023).

## C  TRAINING DETAILS

Our model is optimized using the AdamW optimizer with $\beta_1 = 0.9$ and $\beta_2 = 0.95$, with gradient clipping set to 1.0 and weight decay to 0.1. A cosine learning rate schedule is employed, with a maximum learning rate of 1e-5 and a warmup ratio of 0.008. To accelerate the training, we employ packing (Raffel et al., 2020), where multiple training instances are grouped into a single sequence separated by end-of-sequence tokens, allowing for training on more tokens in each batch.

## D  ADDITIONAL EVALUATION RESULTS

To further illustrate the effectiveness of FUSELLM, we incorporate additional generative benchmarks related to *knowledge-based question-answering*, *reading comprehension*, *content analysis*, *machine translation*, and *theorem application*. The results presented in Table 8 highlight FuseLLM's superiority over all source LLMs across all tasks.

• TriviaQA (Joshi et al., 2017) is a benchmark to evaluate the *knowledge-based question-answering* ability. We conduct a zero-shot evaluation and report the EM accuracy.

• DROP (Dua et al., 2019) is a benchmark to evaluate the *reading comprehension* ability. We conduct a few-shot evaluation with CoT prompts and report the EM accuracy.

• LAMBADA (Paperno et al., 2016) is a benchmark to evaluate the *content analysis* ability. We conduct a zero-shot evaluation and report the EM accuracy.

• IWSLT2017 (Cettolo et al., 2017) is a benchmark to evaluate the *machine translation* ability. We conduct a zero-shot evaluation and report the BLEU (Papineni et al., 2002) score.

• SciBench (Wang et al., 2023a) is a benchmark to evaluate the *theorem application* ability. We conduct a few-shot evaluation with CoT prompts and report the EM accuracy.

| Task | OpenLLaMA | MPT | Llama-2 | Llama-2 CLM | FUSELLM |
|---|---|---|---|---|---|
| TrivialQA | 39.96 | 28.89 | 52.46 | 53.14 (+1.30%) | **54.49** (+3.87%) |
| DROP | 22.31 | 23.54 | 27.25 | 28.51 (+4.62%) | **28.97** (+6.31%) |
| LAMBADA | 70.31 | 70.08 | 73.28 | 73.45 (+0.23%) | **73.72** (+0.60%) |
| IWSLT2017 | 5.51 | 5.49 | 6.48 | **6.91** (+6.64%) | 6.75 (+4.17%) |
| SciBench | 0.68 | 0.88 | 0.14 | 0.94 (+571.43%) | **1.65** (+1078.57%) |

Table 8: Overall results of FUSELLM and baselines in additional generative benchmarks, where percentages indicate the rate of improvement/decrease compared to Llama-2.

## E  FUSELLM VS. PREVIOUS MODEL FUSION METHODS

The motivation behind FuseLLM is to integrate the collective knowledge of multiple LLMs with diverse architectures and pre-training corpora. Consequently, the traditional fusion method of model merging, which demands identical model architectures, is not directly applicable in this context. While the model ensemble technique aggregates predictions from multiple LLMs, the drawback lies in the substantial memory and time costs when maintaining multiple source LLMs during inference. We further compare FUSELLM with an ensemble method for LLMs, LLM-Blender (Jiang et al., 2023), which ranks and com-

| Model | BBH | ME |
|---|---|---|
| OpenLLaMA | 33.87 | **18.11** |
| MPT | 33.38 | 17.26 |
| Llama-2 | 39.70 | 14.63 |
| LLM-Blender (Rank&Fuse) | 24.48 | 0.06 |
| LLM-Blender (Rank) | 37.17 | 17.85 |
| Llama-2 CLM | 40.44 | 14.83 |
| FUSELLM | **41.75** | 15.56 |

Table 9: Comparison of FUSELLM and LLM-Blender.

bines the output texts from multiple LLMs with ranker and fuser models. Specifically, we conduct experiments on the Big-Bench Hard and MultiPL-E benchmarks using the open-source ranker and fuser models. Notably, the CommonSense benchmark, which utilizes perplexity-based evaluation, cannot adapt the LLM-Blender method. The experimental results are shown in Table 9, where LLM-Blender (Rank&Fuse) refers to using the ranker to obtain the top three results and then using the fuser to combine them, and LLM-Blender (Rank) represents simply using the ranker to obtain the top one result. We observed a notable performance deterioration after fusion when employing both the ranker and fuser. This could be attributed to the fuser model's training within the

instruction-tuning context, potentially leading to inadequate generalization to the test tasks. Furthermore, while LLM-Blender (Rank) outperforms the LLM-Blender (Rank&Fuse), it remains inferior to the best-performing source LLM. This suggests the ranker model's inability to discriminate the optimal responses when combining different LLMs efficiently.

## F  INCORPORATING INSTRUCTION-TUNING MODELS WITH FUSELLM

Recall that the proposed FUSELLM involves extracting distribution matrices from multiple distinct source LLMs and continually training the target LLM. Therefore, FUSELLM is also applicable to instruction-tuning models, provided that all corresponding continual-training samples adhere to the instruction-tuning format and mask the instruction part when calculating the training loss.

To confirm this, we conduct new experiments on the fusion of instruction-tuning LLMs. Specifically, we initially fine-tune Llama-2, OpenLLaMA, and MPT using 20k samples from Evol-Instruct (Xu et al., 2023), ShareGPT (Chiang et al., 2023), and Open-Platypus (Lee et al., 2023) datasets, respectively. Consequently, the three source LLMs transitioned into instruction-tuning LLMs. Then, we sample another 5k samples from each of the aforementioned datasets to create a corpus for con-

| Model | Vicuna Benchmark |
|---|---|
| OpenLLaMA ShareGPT | 7.23 |
| MPT Open-Platypus | 6.46 |
| Llama-2 Evol-Instruct | 7.88 |
| Llama-2 Evol-Instruct CLM | 8.03 (+1.90%) |
| FUSELLM | **8.16** (+3.55%) |

Table 10: Results of fusing instruction-tuning models with FUSELLM.

tinual training, specifying Llama-2 Evol-Instruct as the target LLM for knowledge fusion. We assess the instruction-following performance on the Vicuna Benchmark using GPT-4 as an evaluator following Chiang et al. (2023), which gives a score from 1 to 10 for each answer. The results shown in Table 10 demonstrate that FuseLLM surpasses each individual source instruction-tuning LLM, achieving the best performance with GPT-4 judgment.

## G  CAUSE OF PERFORMANCE IMPROVEMENT

To further demonstrate that our performance improvement stems from the integration of knowledge from multiple LLMs rather than solely from continual training, we conduct an evaluation on an alternative corpus, RedPajama (Computer, 2023). To mitigate the impact of different tokenizers, we employ the bits per UTF-8 encoded byte (BPB) metric proposed by Gao et al. (2020), where a smaller value indicates lower perplexity. Then, we compute the percentage of test samples in each domain exhibiting decreased BPB from Llama-2 CLM to FUSELLM and from Llama-2 to OpenLLaMA or MPT. The results in Figure 4 suggest that when FUSELLM outperforms Llama-2 CLM, the performance of OpenLLaMA or MPT typically surpasses that of Llama-2, as evidenced

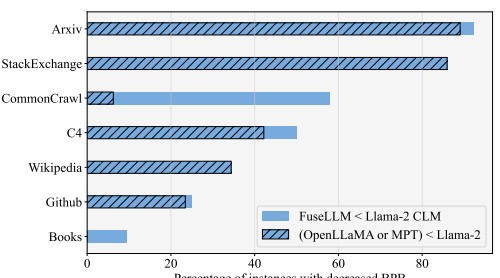

Figure 4: Perplexity comparison on RedPajama. The bars denote the percentage of examples with reduced perplexity when transitioning from CLM to FUSELLM (solid) and from Llama-2 to Open-LLaMA or MPT (slashed).

by the slashed bars in each domain. This phenomenon is particularly pronounced in domains such as Arxiv, StackExchange, Wikipedia, and Github, where it exceeds 95%. This compelling evidence suggests that the performance enhancements achieved by FUSELLM are indeed attributed to the integration of knowledge from multiple LLMs.

## H  WEIGHT FOR LOSS COMBINATION

As an alternative to the static weight for $\lambda$ in Eq. 5, we consider a dynamic design, which is referred to as teacher annealing (Clark et al., 2019b). This technique gradually increases $\lambda$ during the training process, giving preference

| Design | BBH | ME | CS |
|---|---|---|---|
| Dynamic | 41.75 | 15.52 | 64.50 |
| Static | 41.75 (+0.00%) | 15.56 (+0.25%) | 64.56 (+0.09%) |

Table 11: Comparison of different designs for $\lambda$.

to $\mathcal{L}_{\text{Fusion}}$ initially and subsequently shifting the focus to $\mathcal{L}_{\text{CLM}}$. In this experiment, we set the $\lambda$ to increase linearly from 0.7 to 1.0. Table 11 presents a comparison between the static and dynamic designs of $\lambda$, demonstrating that the two approaches yield comparable performance. Therefore, we opt for the static design for simplicity.

## I  USING STRONGEST LLM AS TARGET

In the experiments, we consistently employ Llama-2 as the target LLM to maintain a fixed setup. We further conduct supplementary experiments on code generation tasks using the strongest LLM, OpenLLaMA, as the target LLM. The results are shown in Table 12. We observe a decrease in the performance of OpenLLaMA CLM compared to the source OpenLLaMA and MPT. This decline is attributed to the inclusion of the StarCoder (Li et al., 2023) data in OpenLLaMA's pre-training corpus, whereas there is a limited amount of code data in our continual training corpus. For the same reason, even though FuseLLM showcases improved performance compared to OpenLLaMA CLM, it still lags behind the original OpenLLaMA.

| Task | OpenLLaMA | MPT | Llama-2 | OpenLLaMA CLM | FUSELLM |
|---|---|---|---|---|---|
| C++ | **14.47** | 13.11 | 7.45 | 12.89 | 14.01 |
| Go | 68.20 | 66.96 | 57.02 | 68.48 | **69.32** |
| Java | **14.28** | 13.42 | 10.31 | 12.80 | 13.26 |
| JavaScript | **17.61** | 13.01 | 13.17 | 15.50 | 17.02 |
| PHP | **11.24** | 9.53 | 9.75 | 10.06 | 10.09 |
| Python | 15.96 | **17.24** | 13.85 | 15.16 | 16.06 |
| R | **7.52** | 4.53 | 4.97 | 6.89 | 6.68 |
| Ruby | 10.34 | **12.33** | 10.37 | 9.44 | 10.03 |
| Rust | 6.18 | **8.29** | 6.77 | 5.99 | 5.65 |
| TypeScript | 15.31 | 14.13 | 12.61 | 14.10 | **15.37** |
| Avg. 10 Tasks | **18.11** | 17.26 | 14.63 | 17.13 | 17.75 |

Table 12: Overall results of FUSELLM using OpenLLaMA as the target LLM in code generation evaluations on MultiPL-E (ME).

## J  CASE STUDIES

In Table 13, Table 14, and Table 15, we present case studies to demonstrate how FUSELLM combines the strengths of multiple source LLMs to produce accurate results in different tasks.

| **BBH; Logical Deduction (3 objects);** |
|---|
| **Question:** The following paragraphs each describe a set of three objects arranged in a fixed order. The statements are logically consistent within each paragraph. A fruit stand sells three fruits: oranges, watermelons, and pears. The oranges are the second-most expensive. The pears are more expensive than the watermelons. Options: (A) The oranges are the second-most expensive (B) The watermelons are the second-most expensive (C) The pears are the second-most expensive |
| **Target:** (A) |
| **OpenLLaMA:** Let's think step by step. (1) The oranges are the second-most expensive: "(expensive) ? oranges ? watermelons ? (expensive)". (2) The pears are more expensive than the watermelons: "(expensive) ? oranges pears watermelons (expensive)". (3) Combining (1) and (2) we get the following ordering: "(expensive) oranges pears watermelons (expensive)". According to this ordering, the second-most expensive fruit is the oranges. The oranges are the second-most expensive. So the answer is (A). |
| **MPT:** Let's think step by step. (1) The oranges are the second-most expensive: "(most expensive)? oranges? (least expensive)". (2) The pears are more expensive than the watermelons: "(most expensive)? oranges? pears? (least expensive)". (3) Combining (1) and (2) we get the following ordering: "(most expensive) oranges pears (least expensive)". According to this ordering, the second-most expensive fruit is the oranges. The oranges are the second-most expensive. So the answer is (A). |
| **Llama-2:** Let's think step by step. (1) The oranges are the second-most expensive: "(second-most-expensive) ? oranges ? (most-expensive)". (2) The pears are more expensive than the watermelons: "(second-most-expensive) ? oranges pears (most-expensive)". (3) Combining (1) and (2) we get the following ordering: "(second-most-expensive) oranges pears (most-expensive)". According to this ordering, the second-most expensive fruit is the pears. The pears are the second-most expensive. So the answer is (C). |
| **Llama-2 CLM:** Let's think step by step. (1) The oranges are the second-most expensive: "(second-most) ? oranges ? (most)". (2) The pears are more expensive than the watermelons: "(more) ? pears watermelons (less)". (3) Combining (1) and (2) we get the following ordering: "(second-most) oranges pears (most) watermelons". According to this ordering, the second-most expensive fruit is the pears. The pears are the second-most expensive. So the answer is (C). |
| **FUSELLM:** Let's think step by step. (1) The oranges are the second-most expensive: "(second-most) ? oranges ? (most) ? (least) ? (least) ? (most) ? (second-most)". (2) The pears are more expensive than the watermelons: "(more) ? pears ? (less) ? watermelons ? (most) ? (least) ? (least) ? (most)". (3) Combining (1) and (2) we get the following ordering: "(second-most) oranges pears watermelons (most)". According to this ordering, the second-most expensive fruit is the oranges. The oranges are the second-most expensive. So the answer is (A). |

Table 13: Case studies on the Logical Deduction (3 objects) task.

| BBH; Hyperbaton; |
| --- |
| **Question:** Which sentence has the correct adjective order: Options: (A) repulsive prismlike glass shoe (B) glass repulsive prismlike shoe |
| **Target:** (A) |
| **OpenLLaMA:** Let's think step by step. When there is more than one adjective before a noun, the adjectives need to respect the following order before a noun: "[1. opinion] [2. size] [3. age] [4. shape] [5. color] [6. origin] [7. material] [8. purpose] noun". Option (A): "repulsive prismlike glass shoe". (1) "repulsive" falls into the opinion category. (2) "prismlike" falls into the shape category. (3) "glass" falls into the material category. Option (A) has the following adjective order: [1. opinion] [3. shape] [6. material] (or, in numeric terms, 1 3 6). Because 1 ¡ 3 ¡ 6 is correct, (A) has the correct ordering. Option (B): "glass repulsive prismlike shoe". Option (B) has the following adjective order: [6. material] [1. opinion] [3. shape] (or, in numeric terms, 6 1 3). Because 6 ¡ 1 ¡ 3 is not correct, (B) does not have the correct ordering. So the answer is (A). |
| **MPT:** Let's think step by step. When there is more than one adjective before a noun, the adjectives need to respect the following order before a noun: "[1. opinion] [2. size] [3. age] [4. shape] [5. color] [6. origin] [7. material] [8. purpose] noun". Option (A): "repulsive prismlike glass shoe". (1) "repulsive" falls into the opinion category. (2) "prismlike" falls into the shape category. (3) "glass" falls into the material category. Option (A) has the following adjective order: [1. opinion] [3. material] [2. shape] (or, in numeric terms, 1 3 2 ). Because 1 ¡ 3 ¡ 2 is not correct, (A) does not have the correct ordering. Option (B): "glass repulsive prismlike shoe". Option (B) has the following adjective order: [1. opinion] [3. material] [2. shape] (or, in numeric terms, 1 3 2 ). Because 1 ¡ 3 ¡ 2 is correct, (B) has the correct ordering. So the answer is (B). |
| **Llama-2:** Let's think step by step. When there is more than one adjective before a noun, the adjectives need to respect the following order before a noun: "[1. opinion] [2. size] [3. age] [4. shape] [5. color] [6. origin] [7. material] [8. purpose] noun". Option (A): "repulsive prismlike glass shoe". (1) "repulsive" falls into the opinion category. (2) "prismlike" falls into the shape category. (3) "glass" falls into the material category. Option (A) has the following adjective order: [1. opinion] [2. size] [3. age] [6. origin] [7. material] (or, in numeric terms, 1 2 3 6 7). Because 1 ¡ 2 ¡ 3 ¡ 6 ¡ 7 is not correct, (A) does not have the correct ordering. Option (B): "glass repulsive prismlike shoe". Option (B) has the following adjective order: [1. opinion] [2. size] [3. age] [4. shape] [7. material] (or, in numeric terms, 1 2 3 4 7). Because 1 ¡ 2 ¡ 3 ¡ 4 ¡ 7 is correct, (B) has the correct ordering. So the answer is (B). |
| **Llama-2 CLM:** Let's think step by step. When there is more than one adjective before a noun, the adjectives need to respect the following order before a noun: "[1. opinion] [2. size] [3. age] [4. shape] [5. color] [6. origin] [7. material] [8. purpose] noun". Option (A): "repulsive prismlike glass shoe". (1) "repulsive" falls into the opinion category. (2) "prismlike" falls into the shape category. (3) "glass" falls into the material category. Option (A) has the following adjective order: [1. opinion] [2. size] [3. age] [4. shape] [5. color] [7. material] (or, in numeric terms, 1 2 3 4 5 7). Because 1 ¡ 2 ¡ 3 ¡ 4 ¡ 5 ¡ 7 is correct, (A) has the correct ordering. Option (B): "glass repulsive prismlike shoe". Option (B) has the following adjective order: [7. material] [1. opinion] [2. size] [3. age] [4. shape] (or, in numeric terms, 7 1 2 3 4). Because 7 ¡ 1 ¡ 2 ¡ 3 ¡ 4 is correct, (B) has the correct ordering. So the answer is (B). |
| **FUSELLM:** Let's think step by step. When there is more than one adjective before a noun, the adjectives need to respect the following order before a noun: "[1. opinion] [2. size] [3. age] [4. shape] [5. color] [6. origin] [7. material] [8. purpose] noun". Option (A): "repulsive prismlike glass shoe". (1) "repulsive" falls into the opinion category. (2) "prismlike" falls into the shape category. (3) "glass" falls into the material category. Option (A) has the following adjective order: [1. opinion] [2. size] [3. age] [4. shape] [5. color] [6. origin] [7. material] (or, in numeric terms, 1 2 3 4 5 6 7). Because 1 ¡ 2 ¡ 3 ¡ 4 ¡ 5 ¡ 6 ¡ 7 is correct, (A) has the correct ordering. Option (B): "glass repulsive prismlike shoe". Option (B) has the following adjective order: [1. opinion] [7. material] [2. size] [3. age] [4. shape] [5. color] (or, in numeric terms, 1 7 2 3 4 5). Because 1 ¡ 7 ¡ 2 ¡ 3 ¡ 4 ¡ 5 is not correct, (B) does not have the correct ordering. So the answer is (A). |

Table 14: Case studies on the Hyperbaton task.

| BBH; Disambiguation QA; |
| --- |
| **Question:** In the following sentences, explain the antecedent of the pronoun (which thing the pronoun refers to), or state that it is ambiguous. Sentence: The technician told the customer that he could pay with cash. Options: (A) The technician could pay (B) The customer could pay (C) Ambiguous |
| **Target:** (B) |
| **OpenLLaMA:** Let's think step by step. Here we need to determine who the pronoun "he" might be referring to. There are two possible referents for "he", namely the technician and the customer. The verb "told" might be able to help us determine which one is more likely (if either). Let X be the technician and Y the customer. The sentence is then of the form "X told Y that (X or Y) could pay with cash." Let's consider Y first: "X told Y that Y could pay with cash." This case makes sense, because X might want to inform Y that Y could pay with cash. Now, consider X: "X told Y that X could pay with cash." This case also makes sense, because X might want to inform Y that X could pay with cash. Because both X and Y are possible at the same time, we conclude that the antecedent of the pronoun is ambiguous. So the answer is (C). |
| **MPT:** Let's think step by step. Here we need to determine who the pronoun "he" might be referring to. There are two possible referents for "he", namely the technician and the customer. The verb "told" might be able to help us determine which one is more likely (if either). Let X be the technician and Y the customer. The sentence is then of the form "X told Y that (X or Y) could pay." Let's consider Y first: "X told Y that Y could pay." This case makes sense, because Y might want to pay with cash. Now, consider X: "X told Y that X could pay." This case doesn't make sense, because X would already have the information that X could pay, because it is information about themself. Because in this context, X is the technician and Y is the customer, the answer should be the customer. So the answer is (B). |
| **Llama-2:** Let's think step by step. Here we need to determine who the pronoun "he" might be referring to. There are two possible referents for "he", namely the technician and the customer. The verb "told" might be able to help us determine which one is more likely (if either). Let X be the technician and Y the customer. The sentence is then of the form "X told Y that (X or Y) could do something." Let's consider Y first: "X told Y that Y could do something." This case makes sense, because Y is the one who is being told that they can do something. Now, consider X: "X told Y that X could do something." This case doesn't make sense, because X is the one who is telling Y that they can do something. Because in this context, X is the technician and Y is the customer, the answer should be the technician. So the answer is (A). |
| **Llama-2 CLM:** Let's think step by step. Here we need to determine who the pronoun "he" might be referring to. There are two possible referents for "he", namely the technician and the customer. The verb "told" might be able to help us determine which one is more likely (if either). Let X be the technician and Y the customer. The sentence is then of the form "X told Y that (X or Y) could pay." Let's consider Y first: "X told Y that Y could pay." This case doesn't make sense, because Y is the receiver of the information, so it is now X's turn to pay. Now, consider X: "X told Y that X could pay." This case makes sense, because X is the one who is telling Y that X can pay. Because in this context, X is the technician and Y is the customer, the answer should be the technician. So the answer is (A). |
| **FUSELLM:** Let's think step by step. Here we need to determine who the pronoun "he" might be referring to. There are two possible referents for "he", namely the technician and the customer. The verb "told" might be able to help us determine which one is more likely (if either). Let X be the technician and Y the customer. The sentence is then of the form "X told Y that (X or Y) could pay." Let's consider Y first: "X told Y that Y could pay." This case makes sense, because Y is the customer, and the customer is the one who is paying. Now, consider X: "X told Y that X could pay." This case doesn't make sense, because X is the technician, and the technician is not the one who is paying. Because in this context, X is the technician and Y is the customer, the answer should be the customer. So the answer is (B). |

Table 15: Case studies on the Disambiguation QA task.

