# OpenReview forum: "Knowledge Fusion of Large Language Models"
_ICLR.cc/2024/Conference — ICLR 2024 poster_

### Official Review · Reviewer_jpJv · 2023-10-27

**Soundness:** 4 excellent
**Presentation:** 3 good
**Contribution:** 3 good
**Rating:** 6
**Confidence:** 3

**Summary:**

The paper states that current method for training a LLM from scratch requires a significant cost and may result in redundant capabilities. While current model ensembling method and direct parameters pooling are either computational expensive or impractical due to the difference of model structure. Based on this the paper treats the probabilistic matrix generated by a LLM given a text as the knowledge of LLM, and proposed a framework named FuseLLM to minimize the discrepancy between the probabilistic matrix of the target LLM and the source LLMs. The empirical study of the proposed approach show the strong performance of the method. Besides, the paper also compare FuseLLM to knowledge distillation and the traditional model ensembling / merging methods to further justify the performance of the FuseLLM.

**Strengths:**

1. The propose approach which allows merging knowledge from multiple sources of LLMs is novel and straightforward.
2. The empirical study shows strong performance of the proposed strategy.

**Weaknesses:**

The implementation part is not very well written, I cannot completely understand the specific details toward the mechanism of the alignment and fusion.

**Questions:**

1. For the detailed implementation part, I am a little bit confused. Can you explain in details how MinED works for token alignment, and how MinCE/AvgCE merge different matrices into P_t?

2. As a combined objective is used in FuseLLM (5), what is the choice of $\lambda$ in your experiment? I saw from Appendix E that for dynamic design $\lambda$ is set from 0.7 - 1.0, while 1.0 reduces the (5) to the regular CLM objective. Could you provide a more detailed elaboration on this hyperparameter?

3. In Table 1, interestingly FuseLLM receive significant improvement in several tasks (e.g., Hyperbaton, Web of Lies) from MPT and OpenLLaMA, which is actually expected. While there are still a lot of the tasks like Formal Fallacies, Causal Judgement, and most of the code generation tasks, where significant improvement are not observed. What if the strongest source model is used for continuing training, instead of the Llama-2? For example, OpenLLaMA for code generation.

---

> ### Author Response · Authors · 2023-11-19
>
> We sincerely appreciate the time and effort you put into reviewing our paper. We appreciate your insightful feedback and are pleased that you find our work to be both novel and straightforward. We will carefully address each of your concerns in the following points.
>
> **Q1: Regarding the specific implementation details of alignment and fusion.**
>
> **A1:**
>
> 1）MinED works for token alignment
>
> For an input text x, token alignment involves aligning two distribution matrices from two LLMs. Therefore, the alignment comprises two dimensions: token-wise with respect to text x and distribution-wise with respect to the vocabulary.
>
> In the token dimension, we utilize the dynamic programming approach to recursively minimize the total cost of editing one sequence of tokens to align with another. When the mapped tokens are identical, such as the token "now" in the given example, these tokens are effectively aligned, allowing for the corresponding distributions to align subsequently. However, when the mapped tokens exhibit differences, such as the 'get' and 'gets' tokens in the example, the previous EM method does not align these tokens, resulting in the distributions degenerating to one-hot vectors. In contrast, our proposed MinED method successfully aligns the 'gets' token with the 'get' token, as they exhibit the minimal edit distance in the vocabularies derived from the two LLMs.
>
> Concerning the distribution dimension, the alignment is performed between two vocabularies from different tokenizers of two LLMs. Therefore, for distribution values with identical tokens, such as 'current 0.05' and 'current 0.04', they will be aligned effectively. For distribution values involving different tokens, such as 'immediate 0.04' and 'immediately 0.03', the EM method disregards this value. However, the proposed MinED method maps 'immediately' to 'immediate' due to their minimal edit distance, resulting in the successful alignment of these distribution values.
>
> An example: https://anonymous.4open.science/r/ICLR24_Anonymous-9D08/token_and_vocab_alignment.pdf
>
> We will include the above explanation when updating the paper.
>
> 2）MinCE/AvgCE works for knowledge fusion
>
> Following token alignment, the next crucial problem in knowledge fusion is to merge the distribution matrices of text x from different LLMs and continually train the target LLM. We selected the cross-entropy loss between the distribution matrices and the gold labels as an indicator of the prediction quality of the LLMs. Specifically, we utilized either the minimum cross-entropy scores (MinCE) or a weighted average of the matrices based on cross-entropy scores (AvgCE) to merge the distribution matrices from different LLMs.
>
> **Q2: Regarding the choice of λ in the experiment.**
>
> **A2:** When introducing the training details in Section 4.1 (third paragraph), we specified that FuseLLM utilizes a fixed value of λ set at 0.9. In Appendix E, an alternative approach is discussed, wherein λ dynamically adjusts during training. As the training progresses, λ is linearly increased from 0.7 to 1.0, as further elaborated in the appendix. We will offer a more detailed explanation of this hyperparameter when revising the paper.
>
> **Q3: Regarding the use of the strongest source model for continual training, e.g. OpenLlama for code generation.**
>
> **A3:** In the experiments, we consistently employed Llama-2 as the target model to maintain a fixed setup. In response to your question, we conducted supplementary experiments using OpenLlama as the target LLM for code generation tasks. The results are presented below.
>
> | Models | Cpp | Go | Java | JavaScript | PHP | Python | R | Ruby | Rust | TypeScript | Avg. |
> | :-: | :-: | :-: | :-: | :-: | :-: | :-: | :-: | :-: | :-: | :-: | :-: |
> | MPT | 13.11 | 66.96 | 13.42 | 13.01 | 9.53 | **17.24** | 4.53 | **12.33**| **8.29** | 14.13 | 17.26 |
> | Llama-2 | 7.45 | 57.02 | 10.31 | 13.17 | 9.75 | 13.85 | 4.97 | 10.37 | 6.77 | 12.61 | 14.63 |
> | OpenLlama | **14.47** | 68.20 | **14.28** | **17.61** | **11.24** | 15.96 | **7.52** | 10.34 | 6.18 | 15.31 | **18.11** |
> | OpenLlama CLM | 12.89 | 68.48 | 12.80 | 15.50 | 10.06 | 15.16 | 6.89 | 9.44 | 5.99 | 14.10 | 17.13 |
> | FuseLLM | 14.01 | **69.32** | 13.26 | 17.02 | 10.09 | 16.06 | 6.68 | 10.03 | 5.65 | **15.37** | 17.75 |
>
> From the results, we observe a decrease in the performance of OpenLlama CLM compared to the source OpenLlama and MPT. This decline is attributed to the inclusion of the Starcoder data in OpenLlama's pre-training corpus, whereas there is a limited amount of code data in our continually training corpus. For the same reason, even though FuseLLM showcases improved performance compared to OpenLlama CLM, it still lags behind the original OpenLlama. We will include these results and provide discussions in the revised version.
>
> We sincerely thank the reviewer again for your valuable suggestions. If you have any further questions, please let us know and we will be happy to discuss them with you further.

---

### Official Review · Reviewer_M2ne · 2023-10-28

**Soundness:** 3 good
**Presentation:** 3 good
**Contribution:** 2 fair
**Rating:** 6
**Confidence:** 4

**Summary:**

The paper discusses the challenges and drawbacks of training large language models (LLMs) from scratch and proposes a cost-effective approach called knowledge fusion (FuseLLM). Knowledge fusion aims to merge existing pre-trained LLMs by leveraging their collective knowledge and strengths to create a more powerful single LLM. The study validates this approach using three different LLM architectures and demonstrates improved performance in areas like reasoning, commonsense, and code generation.

**Strengths:**

- The paper proposed a novel fusion approach for LLMs from a probabilistic distribution perspective.
- The experiments are relatively sufficient.
- Then writing and presentation are clear and easy to read.

**Weaknesses:**

- The presented results could not support that the FuseLLM cuts the cost of initial training and has superiority to other fusion strategies. Lack of quantitative comparison (both results and the cost) between other strategies (continual training, ensemble, merging ...) and the proposed FuseLLM. I only find Table 7 that presents the perplexity.
- The results (Table 1, 2 and 3) show that FuseLLM could not outperform original LLMs in some tasks (some in BBH and all the ME). Does it mean that the FuseLLM only work in some domains? Or in another word, does the FuseLLM could only inherit specific capabilities of LLMs? Have the authors studied this phenomenon? Overall, the improvements are not remarkable, will such fusion effort necessary (with a maybe large training cost)?
- The experiments are only based on Llama-2, OpenLLama and MPT and on 7B scale. Why choosing these three types of models? Can the authors give some explanations? We know that the model will emerge powerful capabilities via scaling up. Maybe the fusion effectiveness will drop with the parameter scale grows. Authors may provide a discussion about this argue.
- How do the Avg. values be calculated in the last lines of Table 1,2 and 3? Directly averaging all the values does not make sense due to the disparate data scales of each task.
- Are the Q_t in Eq. 4 and the Q_t in Eq. 2 the same? I guess the Q_t in Eq. 4 should be used another notation.

**Questions:**

See in the weaknesses.

---

> ### Author Response · Authors · 2023-11-19
> **Official Comment by Authors: Part 1**
>
> We sincerely appreciate the time and effort you put into reviewing our paper. We appreciate your insightful feedback and are pleased that you find our work to be both novel and sufficient. We will carefully address each of your concerns in the following points.
>
> **Q1: Regarding the quantitative comparison of results and cost between continual training, ensemble, and merging.**
>
> **A1:**  In Section 4.2, we compared our proposed FuseLLM method with the continual training approach. We did not include a comparison with the model merging technique, as it is not applicable for fusing multiple source LLMs with distinct architectures. Regarding the model ensemble approach, we have supplemented our experiments by incorporating LLM-Blender on BBH and MultiPL-E benchmarks. The LLM-Blender consists of a ranker and a fuser, which initially rank the output texts generated by multiple LLMs and subsequently combine the top-k results. The experimental results are presented below, with LLM-Blender-top-3-rank-fuse denoting the utilization of the ranker to obtain the top-3 results and the fuser to merge them. LLM-Blender-top-1-rank indicates the sole use of the ranker to acquire the top-1 result. We can observe a notable performance deterioration after fusion when employing both the ranker and fuser. This could be attributed to the fuser model's training within the instruction-tuning context, potentially leading to inadequate generalization to the test tasks. Furthermore, while LLM-Blender outperforms the combined ranker and fuser approach when solely utilizing the ranker, it remains inferior to the best-performing source LLM. This suggests the ranker model's inability to efficiently discriminate the optimal responses when combining different LLMs. The new experiments will be incorporated into the revised version.
>
> | Models | BBH | MultiPL-E |
> | :-: | :-: | :-: |
> | OpenLlama | 33.87 | **18.11** |
> | MPT | 33.38 | 17.26 |
> | Llama-2 | 39.70 | 14.63 |
> | LLM-Blender-top-3-rank-fuse | 24.48 | 0.062 |
> | LLM-Blender-top-1-rank | 37.17 | 17.85 |
> | Llama-2 CLM | 40.44 | 14.83 |
> | FuseLLM | **41.75** | 15.56 |
>
> In terms of cost, our approach utilizes only 1.8B tokens and requires 33 hours of training on 8 40G A100 GPUs. This represents a substantial saving in resources and computational costs compared to the alternative of pre-training from scratch. In contrast to continual training, FuseLLM requires only an additional loss calculation for L_{Fusion}, with the incremental computational cost for obtaining representation matrices from multiple source LLMs being minimal, given that model weights are frozen and only a forward process is needed. When contrasted with the model ensemble approach, such as LLM-Blender, which involves training a ranker and a fuser, along with deploying all LLMs, the ranker, and the fuser during inference, FuseLLM simplifies the process by requiring the deployment of a single model. Regarding the model merging technique, while it requires lightweight training for merging weights and involves running a single merged model during inference, it is unsuitable for fusing LLMs with different architectures.
>
> **Q2: Regarding the FuseLLM could not outperform original LLMs in some tasks and only inherit specific capabilities of LLMs.**
>
> **A2:** Given that we assessed the proposed FuseLLM across a diverse set of 42 tasks, including reasoning, commonsense, and code generation, expecting significant improvements on every task for the fused model appears challenging and, in some cases, unrealistic. There are two primary factors for this difficulty, as discussed in Section 4.2. Firstly, certain source LLMs exhibit suboptimal performance compared to the target LLM in specific tasks, inevitably influencing the fusion results. Secondly, the relevance between the continual training dataset and downstream tasks also plays a crucial role in performance. Considering the diverse domains and tasks covered by downstream tasks, considerable effort is required to curate a corpus tailored for continual training.
>
> In addition to the 42 tasks, we have conducted supplementary experiments on five new generative tasks (knowledge-based question-answering, reading comprehension, content analysis, machine translation, and theorem application) during this rebuttal. The experimental results further validate the efficacy of FuseLLM. We will provide a detailed explanation and incorporate these new experiments in the paper update.
>
> | Models | TrivialQA | Drop | Lambada | IWSLT2017 | SciBench |
> | :-: | :-: | :-: | :-: | :-: | :-: |
> | OpenLlama | 39.96 | 22.31| 70.31 | 5.51 | 0.677 |
> | MPT | 28.89 | 23.54 | 70.08 | 5.49 | 0.876 |
> | Llama-2 | 52.46 | 27.25 | 73.28 | 6.48 | 0.137 |
> | Llama-2 CLM | 53.14 | 28.51 | 73.45 | **6.91** | 0.937 |
> | FuseLLM | **54.49** | **28.97** | **73.72** | 6.75 | **1.649** |

---

> ### Author Response · Authors · 2023-11-19
> **Official Comment by Authors: Part 2**
>
> **Q3: Regarding the choosing of Llama-2, OpenLlama, and MPT on a 7B scale.**
>
> **A3:** The motivation behind FuseLLM is to integrate the collective knowledge of multiple LLMs. This collective knowledge manifests in performance variations among source LLMs across diverse inputs and tasks, primarily stemming from disparities in their architecture and pre-training corpora. Among the three LLMs that we have selected, OpenLlama's pre-training corpus comprises Falcon refined-web data and the Starcoder dataset, which differs from Llama-2 in terms of the pre-training corpus. Meanwhile, MPT's pre-training corpus includes S2ORC and The Stack, and it replaces positional encoding with ALiBi and expands the dimension of the intermediate layer in FFN, making it distinct from Llama-2 in both pre-training corpus and architecture.
>
> The reasons for using LLMs at the 7B scale are twofold. Firstly, while our method is applicable to the fusion of larger source LLMs, such as 70+B LLMs, the cost associated with continually training a target large LLM is substantial unless we opt for a smaller target model. Secondly, the availability of choices for source LLMs diminishes as we scale up. Therefore, we conducted experiments on the 7B scale to demonstrate the effectiveness of the proposed fusion method and discussed the use of a larger 13B LLM to distill a smaller 7B LLM in Section 4.5.
>
> Currently, we are conducting experiments to fuse larger LLMs (13B), and we will update the results once the experiment is concluded.
>
> **Q4: Regarding the Avg. values calculating.**
>
> **A4:** Following previous work, such as Llama-2 [1], we employed simple averages for tasks belonging to the same category, such as the 27 tasks in BBH. In response to your question, we supplement results using weighted averages based on the number of samples in each task, and the results are shown below. We will clarify this in the revised version.
>
> | Models | BBH | CommonSense | MultiPL-E |
> | :-: | :-: | :-: | :-: |
> | OpenLlama | 33.73 | 70.24 | **17.94** |
> | MPT | 33.16 | 71.85 | 17.08 |
> | Llama-2 | 39.63 | 73.21 | 14.48 |
> | Llama-2 CLM | 40.42 | 73.41 | 14.69 |
> | FuseLLM | **41.76** | **73.91** | 15.41 |
>
> [1] Hugo Touvron, Louis Martin, Kevin Stone, Peter Albert, Amjad Almahairi, Yasmine Babaei, Nikolay Bashlykov, Soumya Batra, Prajjwal Bhargava, Shruti Bhosale, et al. Llama 2: Open foundation and fine-tuned chat models. arXiv preprint arXiv:2307.09288, 2023.
>
> **Q5: Regarding the O_t in Eq.2 and Q_t in Eq.4.**
>
> **A5:** We would like to clarify that there is no problem with the notation. In Eq.2, the one-hot label matrix is denoted as O_t, while in Eq.4, the output distribution matrix of the target LLM is represented by Q_t.
>
> We sincerely thank the reviewer again for your valuable suggestions! Please let us know if you have any further questions, and we are happy to discuss further.

---

### Official Review · Reviewer_CN5G · 2023-10-30

**Soundness:** 3 good
**Presentation:** 3 good
**Contribution:** 3 good
**Rating:** 8
**Confidence:** 3

**Summary:**

This work introduces a novel training method that utilizes prediction probabilities from LLMs as a teaching signal to tuning an LLM. Unlike conventional teacher-student distillation approaches, this work proposes token alignment to handle vocabulary differences and fusion methods to combine the predictions of multiple LLMs, which means the teaching signal is a mixed prediction distribution.

**Strengths:**

Training LLMs is expensive, making the motivation to merge existing LLMs valuable.

The paper is well-written and easy to follow, and the experiment setup and ablation studies are solid and insightful.

**Weaknesses:**

I think the proposed method presents a promising training algorithm for leveraging multiple LLMs as teachers. However, it's better to have insights about what the student LLM has learned. The best empirical fusion method, Min Cross-entropy, utilizes token-level probability maximization across multiple teachers as the teaching signal. This teaching signal is not intuitive. Considering Hinton's dark knowledge theory, why does token-level max pooling outperforms average pooling?

**Questions:**

To my understanding, teacher models are frozen during the training process, allowing us to infer and store their prediction distributions before training. This enables the utilization of larger models, such as employing 70+B LLMs as teachers. Since teacher inference occurs only once, the additional cost should not be significant.

Regarding the Token alignment, I am still unsure the method after reading Appendix A. Can you provide an example or some explanations?

---

> ### Author Response · Authors · 2023-11-19
>
> We sincerely appreciate the time and effort you put into reviewing our paper. We appreciate your insightful feedback and we are thankful for your recognition of our work's novelty and significance. In the subsequent points, we will carefully address each of your concerns.
>
> **Q1: Regarding the insights about what the student LLM has learned.**
>
> **A1:** In our FuseLLM approach, the fusion function is crafted to consistently learn from the best prediction (or combination) among multiple source LLMs for a given text input. This ensures that the target LLM consistently integrates the most robust capabilities of the LLMs involved. We will incorporate this clarification in the paper update.
>
> **Q2: Regarding the token-level max pooling outperforms average pooling.**
>
> **A2:**  In our implementation, FuseLLM chooses the distribution from an optimal source LLM at the sequence level and performs distillation, akin to sequence-level max pooling rather than token-level max pooling. Given that generative tasks require coherence among tokens, the sequence-level max pooling approach effectively maintains sequential coherence, making it a more reasonable choice. We will provide further clarification on this point in the revised version.
>
> **Q3: Regarding the utilization of larger models, such as employing 70+B LLMs as teachers.**
>
> **A3:** Yes, our method is applicable to the fusion of larger source LLMs, such as 70+B LLMs. While these larger LLMs only require a forward process and their weights are frozen when obtaining the distribution, the cost associated with continually training the target large LLM remains significant unless we opt for a smaller target model as multi-teacher knowledge distillation. Additionally, the availability of choices for source LLMs diminishes as we move to larger scales, such as 70+B LLMs. Therefore, we conducted experiments on the 7B scale to showcase the effectiveness of the proposed fusion method and discussed the use of a larger 13B LLM to distill a smaller 7B LLM in Section 4.5.
>
> Currently, we are conducting experiments to fuse larger LLMs (13B), and we will update the results once the experiment is concluded.
>
> **Q4: Regarding the token alignment.**
>
> **A4:**  For an input text x, token alignment involves aligning two distribution matrices from two LLMs. Therefore, the alignment comprises two dimensions: token-wise with respect to text x and distribution-wise with respect to the vocabulary.
>
> In the token dimension, we utilize the dynamic programming approach to recursively minimize the total cost of editing one sequence of tokens to align with another. When the mapped tokens are identical, such as the token "now" in the given example, these tokens are effectively aligned, allowing for the corresponding distributions to align subsequently. However, when the mapped tokens exhibit differences, such as the 'get' and 'gets' tokens in the example, the previous EM method does not align these tokens, resulting in the distributions degenerating to one-hot vectors. In contrast, our proposed MinED method successfully aligns the 'gets' token with the 'get' token, as they exhibit the minimal edit distance in the vocabularies derived from the two LLMs.
>
> Concerning the distribution dimension, the alignment is performed between two vocabularies from different tokenizers of two LLMs. Therefore, for distribution values with identical tokens, such as 'current 0.05' and 'current 0.04', they will be aligned effectively. For distribution values involving different tokens, such as 'immediate 0.04' and 'immediately 0.03', the EM method disregards this value. However, the proposed MinED method maps 'immediately' to 'immediate' due to their minimal edit distance, resulting in the successful alignment of these distribution values.
>
> An example of token alignment: https://anonymous.4open.science/r/ICLR24_Anonymous-9D08/token_and_vocab_alignment.pdf
>
> We will include the above explanation when updating the paper.
>
> We sincerely thank the reviewer again for your valuable suggestions! Please let us know if you have any further questions, and we are happy to discuss further.

---

### Official Review · Reviewer_fXfT · 2023-11-01

**Soundness:** 3 good
**Presentation:** 4 excellent
**Contribution:** 3 good
**Rating:** 6
**Confidence:** 4

**Summary:**

This paper proposes to fuse the power of multiple pretrained LLMs (LLaMA2-7B, MPT-7B, and OpenLLaMA-7B), via a continual pretraining on top of LLaMA2 with the 1 Million-document MiniPile corpus, where the training objective consists of a weighted combination of a normal causal LM loss and a multi-teacher knowledge distillation loss. The result model FuseLLM performs better than LLaMA2 on reasoning (BBH), commonsense, and code generation (MultiPL-E) benchmarks.

**Strengths:**

* This paper is well-written and easy-to-understand, and the details are clearly presented.

**Weaknesses:**

* The proposed approach relying on continual pretraining with a small pretraining corpus (MiniPile) looks limited to fusing the knowledge of the raw pretrained version of LLMs, but not applicable to "-Chat" or "-Instruct" models (e.g., Vicuna, Alpaca, MPT-instruct, ChatGPT, etc.)

* All the benchmarks in this paper are classification tasks. It is unclear if the FuseLLM would practically produce better text outputs on instructions, such as real-world human instructions, e.g., Alpaca (https://raw.githubusercontent.com/tatsu-lab/stanford_alpaca/main/alpaca_data.json) and some generative tasks in Big-Bench, e.g., word_unscrambling.

*  The performance improvement from the proposed fusion approach is not significant enough. Specifically,\
      on reasoning tasks: 39.70 (LLaMA2) -> 40.44 (LLaMA2 + MiniPile) -> 41.78 (LLaMA2 + MiniPile  + model fusion)\
      on commonsense tasks: 63.76 (LLaMA2) -> 63.86 (LLaMA2 + MiniPile) -> 64.56 (LLaMA2 + MiniPile + model fusion)\
      on code generation tasks: 14.63 (LLaMA2) -> 14.83 (LLaMA2 + MiniPile) -> 15.56 (LLaMA2 +MiniPile + model fusion)\
Given the MiniPile continual pretraining is not a key contribution of this paper, the actual accuracy improvement brought by the proposed model fusion on commonsense and code generation is actually smaller than 1 percent.

* The evaluation of this paper doesn't include the comparison of FuseLLM with any of the previous model fusion techniques, e.g., LLM-Blender (Jiang et al.) mentioned in the "related work" section. Notably, the LLM-blender also doesn't require heavy continual training on a 7B large model, which is not discussed.

**Questions:**

See "Weaknesses".

---

> ### Author Response · Authors · 2023-11-19
> **Official Comment by Authors: Part 1**
>
> We sincerely appreciate the time and effort you put into reviewing our paper. We will carefully address each of your concerns in the following points.
>
> **Q1: Regarding the application to “-Chat” or “-Instruct” models.**
>
> **A1:** Recall that the proposed FuseLLM method involves extracting distribution matrices from multiple distinct source LLMs and continually training the target LLM. Therefore, our approach is applicable to both '-Chat' and '-Instruct' LLMs, provided that all corresponding continual-training samples adhere to the instruction-tuning format. It is also essential to mask the instruction part when calculating the training loss. To confirm this, we conducted new experiments on the fusion of '-Instruct' LLMs.
>
> Specifically, we initiated fine-tuning on Llama-2, OpenLlama, and MPT using 20K samples from Evol-Instruct, ShareGPT, and Open-Platypus datasets, respectively. Consequently, the three source LLMs transitioned into '-Instruct' LLMs. Then, we sampled another 5K samples from each of the aforementioned datasets to create a corpus for continual training, designating Llama-2 Evol-Instruct as the target LLM for model fusion. We assessed the instruction-following performance on the Vicuna Benchmark using the GPT-4 scoring standard [1], where a higher score indicates superior performance. The results presented in the following table show that FuseLLM surpasses each individual source '-Instruct' LLM, achieving the best performance.   We will include this experiments when updating the paper.
>
> [1] Vicuna: An open-source chatbot impressing gpt-4 with 90%* chatgpt quality, March 2023. URL https://vicuna.lmsys.org.
>
> | Models | GPT-4 Score |
> | :-: | :-: |
> | OpenLlama ShareGPT | 7.23 |
> | MPT Open-Platypus | 6.46 |
> | Llama-2 Evol-Instruct | 7.88 |
> | Llama-2 Evol-Instruct CLM | 8.03 |
> | FuseLLM | **8.16** |
>
> **Q2: Regarding the generative tasks.**
>
> **A2:** We would like to emphasize that our experiments have already included a range of generative tasks, including Dyck Languages, Multistep Arithmetic Two, Object Counting, and Word Sorting in the BBH, as well as all tasks within the MultiPL-E. In response to your request, we are pleased to incorporate additional generative benchmarks related to knowledge-based question-answering (TriviaQA), reading comprehension (Drop), content analysis (Lambada), machine translation (IWSLT2017), and theorem application (SciBench). The results presented in the following table highlight FuseLLM's superiority over all source LLMs across all tasks. We will include these additional experiments in the revised version.
>
> | Models | TrivialQA | Drop | Lambada | IWSLT2017 | SciBench |
> | :-: | :-: | :-: | :-: | :-: | :-: |
> | OpenLlama | 39.96 | 22.31| 70.31 | 5.51 | 0.677 |
> | MPT | 28.89 | 23.54 | 70.08 | 5.49 | 0.876 |
> | Llama-2 | 52.46 | 27.25 | 73.28 | 6.48 | 0.137 |
> | Llama-2 CLM | 53.14 | 28.51 | 73.45 | **6.91** | 0.937 |
> | FuseLLM | **54.49** | **28.97** | **73.72** | 6.75 | **1.649** |
>
> **Q3: Regarding the not significant performance improvement.**
>
> **A3:** Given that we assessed the proposed FuseLLM across a diverse set of 42 tasks, including reasoning, commonsense, and code generation, expecting significant improvements on every task for the fused model appears challenging and, in some cases, unrealistic. There are two primary factors for this difficulty, as discussed in Section 4.2. Firstly, certain source LLMs exhibit suboptimal performance compared to the target LLM in specific tasks, inevitably influencing the fusion results. Secondly, the relevance between the continual training dataset and downstream tasks also plays a crucial role in performance. Considering the diverse domains and tasks covered by downstream tasks, considerable effort is required to curate a corpus tailored for better continual training.
>
> In addition to the 42 tasks, we have conducted supplementary experiments on five new generation tasks (knowledge-based question-answering, reading comprehension, content analysis, machine translation, and theorem application) during this rebuttal (refer to Q2). The experimental results further validate the efficacy of FuseLLM. We will provide a detailed explanation and incorporate these new experiments in the paper update.

---

> ### Author Response · Authors · 2023-11-19
> **Official Comment by Authors: Part 2**
>
> **Q4: Regarding the comparison to previous model fusion techniques.**
>
> **A4:** The motivation behind FuseLLM is to integrate the collective knowledge of multiple LLMs with diverse architectures and pre-training corpora. **Consequently, the traditional fusion method of model merging, which demands identical model architectures, is not directly applicable in this context. While the ensemble technique performs a weighted average of prediction distributions from multiple LLMs, the drawback lies in the substantial memory and time costs when maintaining multiple source LLMs during inference.** Moreover, in the context of generative models that operate through step-by-step inference, the cost associated with distribution averaging at each step is considerable. A relatively feasible ensemble approach, such as LLM-Blender, involves combining the output texts of multiple LLMs. However, this approach requires additional training for a ranker and a fuser, demanding a substantial amount of high-quality labeled data.
>
> For a fair comparison with model merging and model ensemble methods, in Section 4.6 we simulated fusion scenarios where the same LLM underwent continual training on different corpora. To further compare our method with LLM-Blender, we performed new experiments on the BBH and MultiPL-E benchmarks using the open-source ranker and fuser models of LLM-Blender. Notably, the CommonSense benchmark, which utilizes perplexity-based evaluation, cannot accommodate LLM-Blender. The experimental results are presented below, where LLM-Blender-top-3-rank-fuse refers to using the ranker to obtain the top 3 results and then using the fuser to combine them, and LLM-Blender-top-1-rank represents simply using the ranker to obtain the top 1 result. We observed a notable performance deterioration after fusion when employing both the ranker and fuser. This could be attributed to the fuser model's training within the instruction-tuning context, potentially leading to inadequate generalization to the test tasks. Furthermore, while LLM-Blender outperforms the combined ranker and fuser approach when solely utilizing the ranker, it remains inferior to the best-performing source LLM. This suggests the ranker model's inability to efficiently discriminate the optimal responses when combining different LLMs. The new experiments will be incorporated into the revised version.
>
> | Models | BBH | MultiPL-E |
> | :-: | :-: | :-: |
> | OpenLlama | 33.87 | **18.11** |
> | MPT | 33.38 | 17.26 |
> | Llama-2 | 39.70 | 14.63 |
> | LLM-Blender-top-3-rank-fuse | 24.48 | 0.06 |
> | LLM-Blender-top-1-rank | 37.17 | 17.85 |
> | Llama-2 CLM | 40.44 | 14.83 |
> | FuseLLM | **41.75** | 15.56 |
>
> **Q5: Regarding the discussion of LLM-blender which does not require heavy continual training on a 7B large model.**
>
> **A5:** LLM-Blender operates by employing a ranker to assess and rank the outputs from multiple LLMs, while a fuser combines the ranked outputs to generate the final response. Consequently, LLM-Blender demands a significant quantity of high-quality labeled data for training both the ranker and the fuser. During the inference process, deploying all source LLMs, along with the ranker and fuser models, substantially increases the overall inference cost. Moreover, since the fuser takes the concatenation of the top-k outputs from the source LLMs as input, truncation may be necessary to accommodate the fuser's maximum length limitation. This truncation could potentially lead to a decline in performance.
>
> We sincerely thank the reviewer again for your valuable suggestions. If you have any further questions, please let us know and we will be happy to discuss them with you further.

---

> ### Author Response · Authors · 2023-11-22
> **Look forward to engaging in further discussions with you**
>
> Dear Reviewer,
>
> Thank you for taking the time to review our submission and for your valuable feedback.
>
> We have endeavored to address your questions systematically, providing detailed clarifications and supplementary experiments. We trust that our responses sufficiently address your concerns. Given the significant difference in your rating compared to other reviewers, potentially influencing the final decision, we sincerely request that you review our responses and inform us of any further concerns. Thank you for your consideration.
>
> Best regards,
>
> The Authors

---

> > ### Comment · Reviewer_fXfT · 2023-12-05
> >
> > Thanks for the explanations and the additional experiments! They indeed address many of my concerns. I've updated my rating to 6!

---

### Author Response · Authors · 2023-11-22
**Looking Forward to the Opportunity for Further Discussion**

Dear Reviewers,

We sincerely appreciate the time and effort you've devoted to reviewing our work. We understand that your schedule may be quite busy. As the authors-reviewer discussion phase draws to a close, we kindly request your attention to our responses. Our aim is to gain insights into whether our responses effectively address your concerns and to ascertain if there are any additional questions or points you would like to discuss.

We look forward to the opportunity for further discussion with you. Thank you for your thoughtful consideration.

Best regards,

The Authors

---

### Meta-Review · Area_Chair_wthZ · 2023-12-08

**Metareview:**

This papers presents a technique to distill multiple LLMs into a single model. The method can be contrasted with ensembling (which keeps models separate) and merging (which requires models to be the same architecture). Specifically, the per-token probability distributions are aligned and combined (via one of two different strategies) during continued pre-training. The method outperforms simply performing continued pre-training with the ground-truth labels and compares favorably to ensembling or merging.

**Justification For Why Not Higher Score:**

While the proposed method is effective and provides a new way of combining models, it does not provide dramatic gains compared to various baselines. In addition, reviewers all recommended acceptance but all reviewers pointed out various weaknesses.

**Justification For Why Not Lower Score:**

Reviewers all agreed on acceptance.

---

### Decision · Program_Chairs · 2024-01-16

Accept (poster)